# Genetic Characterization of Antimicrobial-Resistant *Escherichia coli* Isolated from a Mixed-Use Watershed in Northeast Georgia, USA

**DOI:** 10.3390/ijerph16193761

**Published:** 2019-10-07

**Authors:** Sohyun Cho, Hoang Anh Thi Nguyen, Jacob M. McDonald, Tiffanie A. Woodley, Lari M. Hiott, John B. Barrett, Charlene R. Jackson, Jonathan G. Frye

**Affiliations:** 1Department of Microbiology, University of Georgia, Athens, GA 30602, USA; sohyun.cho25@uga.edu (S.C.); anh.nguyen171@gmail.com (H.A.T.N.); 2(Present) Houston Methodist Research Institute, Houston, TX 77030, USA; 3Warnell School of Forestry and Natural Resources, University of Georgia, Athens, GA 30602, USA; jmcdon@uga.edu; 4Southeast Coast Network, National Park Service, Athens, GA 30605, USA; 5Bacterial Epidemiology and Antimicrobial Resistance Research Unit, United States Department of Agriculture, Agricultural Research Service, U.S. National Poultry Research Center, Athens, GA 30605, USA; tiffanie.woodley@usda.gov (T.A.W.); lari.hiott@usda.gov (L.M.H.); bennybarrett09@gmail.com (J.B.B.); charlene.jackson@usda.gov (C.R.J.)

**Keywords:** *E. coli*, surface water, antimicrobial resistance, extended spectrum β-lactamase (ESBL), ST131

## Abstract

In order to determine the role of surface water in the development and spread of antibiotic-resistant (AR) bacteria, water samples were collected quarterly from 2015 to 2016 from a mixed-use watershed in Georgia. In our previous study, 496 *Escherichia coli* were isolated from surface water, out of which, 34 isolates were resistant to antimicrobials. For the current study, these 34 AR *E. coli* were characterized using pulsed-field gel electrophoresis, AR gene detection, plasmid replicon typing, class I integron detection, and multi-locus sequence typing. Genes were identified as conferring resistance to azithromycin (*mph*(A)); β-lactams (*bla*_CMY_, *bla*_CTX_, *bla*_TEM_); chloramphenicol (*floR*); streptomycin (*strA*, *strB*); sulfisoxazole (*sul1*, *sul2*); tetracycline (*tetA*, *tetB*, *tetC*); and trimethoprim/sulfamethoxazole (*dhfr5*, *dhfr12*). Five ciprofloxacin- and/or nalidixic-resistant isolates contained point mutations in *gyrA* and/or *parC*. Most of the isolates (n = 28) carried plasmids and three were positive for class I integrons. Twenty-nine sequence types (ST) were detected, including three epidemic urinary-tract-infection-associated ST131 isolates. One of the ST131 *E. coli* isolates exhibited an extended-spectrum β-lactamase (ESBL) phenotype and carried *bla*_CTX-M-15_ and *bla*_TEM-1_. To our knowledge, this is the first study on the emergence of an ESBL-producing *E. coli* ST131 from environmental water in the USA, which poses a potential risk to human health through the recreational, agricultural, or municipal use of this natural resource. This study identified *E. coli* with AR mechanisms to commonly used antimicrobials and carrying mobile genetic elements, which could transfer AR genes to other bacteria in the aquatic environment.

## 1. Introduction

Antibiotic-resistant (AR) pathogens are a significant public health concern as infections caused by these pathogens may result in prolonged hospital stays, higher medical costs, and increased rates of morbidity and mortality [1]. Although *Escherichia coli* are usually commensal bacteria in the gastrointestinal tract of humans and animals, they can cause intestinal and extra-intestinal infections, such as gastroenteritis, urinary tract infection, meningitis, and sepsis [2]. Since *E. coli* can readily acquire resistance and are shown to carry resistance to antimicrobial drugs used in human and veterinary medicine, the monitoring of AR *E. coli* in all settings is necessary to understand their impact associated with AR *E. coli* infections [1,2].

There is a growing recognition of the environment as an important factor in the emergence, transmission, and persistence of AR bacteria in light of the One Health perspective, which accepts that the health of humans, animals, and the environment are closely related [3]. The aquatic environment is a hotspot for the development of AR bacteria. Surface and sub-surface waters receive bacteria harboring AR genes from human and animal wastes, as well as antibiotic residues from healthcare, industrial, and agricultural activities. Surface water can also be a vector for the transfer and spread of AR genes to pathogens that humans are exposed to through drinking water, irrigation, or recreational activities [4]. Since AR bacteria and AR genes in the environment are poorly understood in comparison to those in humans, animals, and in the clinical setting, it is important to enhance our limited knowledge of AR bacteria in aquatic environments and develop an understanding of the molecular characteristics of the AR determinants [3].

Our recent work demonstrated the prevalence and AR of *E. coli* from surface water of the Upper Oconee watershed, a mixed-use watershed in northeast Georgia, USA, over a two-year period [5]. The study showed that *E. coli* are ubiquitous in surface water, and are resistant to antimicrobials commonly used in human and veterinary medicine. In the present study, the genetic basis of AR in these isolates was characterized and the association of the AR determinants with mobile genetic elements were investigated. The goals of this study were to improve understanding of AR *E. coli* in surface waters and evaluate the potential role of surface waters as a reservoir of, and potential vehicle for, the spread of AR. 

## 2. Materials and Methods 

### 2.1. E. coli Isolates

The AR *E. coli* isolates (n = 34) from a previous study were selected for characterization [5]. Isolates were from 24 sampling sites within the Upper Oconee watershed near Athens, GA (Figure 1), and exhibited resistance to at least one of these antimicrobials: amoxicillin-clavulanic acid, ampicillin, azithromycin, cefoxitin, ceftiofur, ceftriaxone, chloramphenicol, ciprofloxacin, gentamicin, nalidixic acid, streptomycin, sulfisoxazole, tetracycline, or trimethoprim-sulfamethoxazole [5].

### 2.2. PCR Detection of AR Genes and Sequence Analysis

Isolates were tested for the presence of AR genes to which they exhibited phenotypic resistance. Resistant isolates were tested using PCR for genes encoding resistance to: β-lactams, tetracycline, trimethoprim/sulfamethoxazole, sulfisoxazole, chloramphenicol, aminoglycosides, and azithromycin (Table 1) [6,7,8,9,10,11,12]. PCR assays were performed as previously described in the references using whole-cell templates prepared by suspending a single bacterial colony in 200 μL of sterile water. Taq DNA polymerase and PCR nucleotide mix used for PCR assays were purchased from Roche Diagnostics (Indianapolis, IN, USA). PCR conditions for *bla*_CMY_ were denaturation at 95 °C for 15 min, 30 cycles of denaturation at 94 °C for 1 min, annealing at 61 °C for 1 min, and elongation at 72 °C for 1 min, with a final extension at 72 °C for 10 min. PCR conditions for the amplification of *bla*_CTX-M_ were similar to *bla*_CMY_ except the annealing temperature was 54 °C. Negative and positive controls were included in all PCR assays. Amplified PCR products were analyzed using electrophoresis on a 2% agarose gel and visualized by staining with ethidium bromide.

Quinolone and fluoroquinolone resistant isolates were screened for mutations in the quinolone-resistance-determining regions (QRDRs) of *gyrA* and *parC* (Table 1) [13,14]. PCR products of the genes were purified using a Qiaquick PCR purification kit (Qiagen, Germantown, MD, USA) according to the manufacturer’s directions and used as templates in the sequencing reactions. Sequencing was prepared with 10 µL of water, 8 µL of BigDye Ready Reaction Mix (Applied Biosystems, Foster City, CA, USA), 1 µL of 3.2 pmol forward or reverse primer, and 1 µL of the purified PCR product. Sequencing was performed with the ABI Prism 3130xl Genetic Analyzer (Applied Biosystems) following the manufacturer’s instructions. Sequence similarity was determined by comparison with the GenBank database using BLAST (www.ncbi.nlm.nih.gov).

### 2.3. Phenotypic and Genotypic Detection of ESBL

Isolates resistant to ceftriaxone (minimum inhibitory concentration (MIC) ≥ 4 mg/L) were considered potential producers of extended-spectrum β-lactamase (ESBL) and selected for further testing. They were assayed phenotypically using cefotaxime/clavulanic acid and ceftazidime/clavulanic acid, and classified as ESBL producers if there was at least a three two-fold decrease of the MIC for ceftazidime or cefotaxime in combination with clavulanic acid compared to the MIC when tested without clavulanic acid [15]. These isolates were genetically characterized via amplification of β-lactamase genes (Table 1). The PCR products were sequenced using the ABI Prism 3130x Genetic Analyzer and analyzed using BLAST, as described in Section 2.2.

### 2.4. Pulsed-Field Gel Electrophoresis (PFGE)

PFGE was performed as previously described [16]. In brief, plugs were prepared by embedding bacterial genomic DNA in 1.0% Seakem Gold agarose (BioWhittaker Molecular Applications, Rockland, ME, USA) and digested with 10 U of *Xba*I (Roche Molecular Biochemicals, Indianapolis, USA). Digested DNA was separated via electrophoresis using the CHEF-DRII PFGE system (Bio-Rad, Hercules, CA, USA) in a 0.5x TBE buffer at 6 V for 19 h with ramped pulse times of 2.16 to 54.17 s. Cluster analysis was generated in BioNumerics (Applied Maths, Austin, TX, USA) using the Dice coefficient and the unweighted pair group method of arithmetic averages (UPGMA), with a 1.5% optimization and 1.5% tolerance [17].

### 2.5. Multilocus Sequence Typing (MLST)

MLST was performed using seven housekeeping genes (*adk*, *fumC*, *gyrB*, *icd*, *mdh*, *purA*, and *recA*), as previously described [18]. PCR products were sequenced using the ABI Prism 3130x Genetic Analyzer (Applied Biosystems), as described in Section 2.2. Sequences were analyzed using BioNumerics (Applied Maths) and sequence types (STs) were assigned through the Center for Genomic Epidemiology [19].

### 2.6. Replicon Typing and Integron Analysis

A PCR-based replicon typing (PBRT) kit (Diatheva, Fano, Italy) was used to determine the presence of 28 plasmid replicons (HI1, HI2, I1-α, M, N, I2, B/O, FIB, FIA, W, L, P, X3, I1-γ, T, A/C, FIIS, U, X1, R, FIIK, Y, X2, FIC, K, HIB-M, FIB-M, and FII) according to the manufacturer’s directions.

The class 1 integron was detected via amplification of the conserved segment of the integrase gene, *intI*, as previously described (Table 1) [20]. PCR conditions consisted of 30 cycles of denaturation at 94 °C for 1 min, annealing at 63 °C for 1 min, elongation at 72 °C for 2 min, and a final extension at 72 °C for 10 min. PCR products were analyzed using electrophoresis, as described in Section 2.2.

## 3. Results

### 3.1. AR Genes 

Fifteen resistance genes were detected among the *E. coli* isolates (Table 2, Appendix A
Appendix A). Three tetracycline resistance genes (*tetA*, *tetB*, *tetC*) were detected in 96% (25/26) of the tetracycline resistant isolates, with *tetA* being detected most often (16/26; 61.5%), followed by *tetB* (9/26; 34.6%) and *tetC* (1/26; 3.8%). Seven of the streptomycin-resistant isolates contained both *strA* and *strB* (7/8; 87.5%) while the remaining streptomycin-resistant isolate had *aadA1* (1/8; 12.5%). Approximately 82% (9/11) of the ampicillin-resistant isolates had *bla*_TEM_. The two isolates that were resistant to third-generation cephalosporins contained either *bla*_CMY_ or *bla*_CTX_. Two sulfisoxazole-resistance genes, *sul1* and *sul2*, were detected in 12.5% (1/8) and 87.5% (7/8) of the sulfisoxazole-resistant isolates, respectively. In the three trimethoprim/sulfamethoxazole-resistant isolates, *dhfr5* (2/3) or *dhfr12* (1/3) were detected. Detection of *mph*(A) (2/2) and *floR* (1/1) was observed in azithromycin-resistant and chloramphenicol-resistant isolates, respectively. Nine resistance genes, *aac*(3)*-Iva*, *aacC2*, *aadA2*, *cat1*, *cat2*, *dhfr1*, *dhfr13*, *tetG*, and *tetM*, were not detected in this study.

Analysis of the QRDRs of *gyrA* and *parC* was performed on the five *E. coli* isolates resistant to nalidixic acid. All five isolates contained a mutation of serine-83 to leucine in *gyrA*; one isolate also had aspartic acid-87 mutated to asparagine (Table 3). Additional *gyrA* mutations were seen outside the QRDR, including a lysine-162 to glutamine mutation and a valine-37 to leucine mutation. Of the five nalidixic-acid-resistant isolates, one was also resistant to ciprofloxacin and had a mutation of serine-80 to isoleucine in the QRDR of *parC*. Three isolates had single mutations in *parC* outside the QRDR, two isolates with a mutation of lysine-247 to glutamic acid and one with a mutation of alanine-192 to glycine (Table 3). The remaining nalidixic-acid-resistant isolate did not carry any *parC* mutation.

### 3.2. ESBL Detection

The results of the phenotypic assay confirmed the presence of one ESBL-producing *E. coli*. Upon the amplification and sequencing of β-lactamase genes, the ESBL-producing *E. coli* was positive for *bla*_CTX-M-15_ and *bla*_TEM-1_, while the non-ESBL producer was positive for *bla*_CMY-2_ and *bla*_TEM-1_ (Table 2 and Appendix A
Appendix A). 

### 3.3. Molecular Characteristics

PFGE revealed only two isolates with indistinguishable PFGE patterns; however, they differed in other characteristics, including AR patterns and the plasmid replicons detected (Figure 2). *E. coli* isolates were previously assigned phylogenetic groups [5], and the result is included in the dendrogram for comparison with PFGE patterns and other data collected in the current study (Figure 2). 

Among the 34 AR *E. coli*, 29 STs were identified, including two new STs (Figure 2, Appendix A
Appendix A). Four STs, ST10 (n = 2), ST58 (n = 2), ST69 (n = 2), and ST131 (n = 3) were identified more than once. The four sets of the identical STs belonged to the same phylogenetic groups—groups C, B1, E, and B2, respectively—but demonstrated different AR profiles and PFGE patterns. 

### 3.4. Mobile Genetic Elements

Most of the *E. coli* isolates (28/34; 82.4%) were positive for one or more plasmid replicons (Figure 2, Appendix A
Appendix A). Out of the 28 replicons tested using PCR, 11 types (A/C, FIA, FIB, FIC, FII, HI1, I1α, P, R, X1, Y) were identified, with FIB being the most common, carried by about half of the resistant *E. coli* isolates (n = 16). FII was the second-most common replicon (n = 11), followed by FIA (n = 4), HI1 (n = 4), and I1α (n = 3). P, A/C, and Y were present in two isolates each, while X1, R, and FIC were represented only once. Class I integron (*intI1*) was detected in three isolates, all of which carried plasmid replicons as well. 

## 4. Discussion

The Upper Oconee watershed in Northeast Georgia, USA, is a mixed-use watershed including relatively pristine headwater streams, as well as streams influenced by agricultural runoff and contaminated effluents from wastewater treatment plants, failing septic systems, and sewer line leaks. This mixture of relatively pristine and highly developed land use makes this watershed a useful proxy for other similar-sized watersheds in the southeastern USA. This study determined the molecular characteristics of AR determinants and mobile genetic elements (MGEs) in this watershed to improve our understanding of the watershed as a catchment for AR, an exchange point of AR genes, and a potential source of exposure for humans and animals to AR bacteria. 

This study revealed 17 genes in the 34 *E. coli* isolates responsible for their phenotypic resistance to the antimicrobials tested. Tetracycline resistance was predominant in the isolates, with *tetA* most frequently detected. When compared to studies conducted in different parts of the world, our findings show similar results with *tetA* being detected most often in *E. coli* from aquatic environments; however, *tetM* was not detected here but was frequently detected in those studies [21,22,23]. Ampicillin and sulfisoxazole resistance could be mostly attributed to *bla*_TEM_ and *sul2*, respectively. Similar prevalence of AR genes was seen in *E. coli* isolates from other aquatic environments [21,22,23]. However, unlike those previous findings in which *cat1*, *aadA*, and *dhfr1* and *dhfr7* were responsible for chloramphenicol, streptomycin, and trimethoprim resistance, respectively, our findings suggested *floR*, *strA* and *strB*, and *dhfr5* were responsible for resistance to those antimicrobials [21,22,23]. A plasmid-borne azithromycin resistance gene, *mph*(A), was detected in two azithromycin-resistant isolates. A study conducted by Nguyen et al. reported that *mph*(A) was the most common macrolide resistance gene among commensal and clinical isolates of *E. coli* from five countries on four continents [12].

Quinolone and fluoroquinolone resistances are commonly mediated by mutations in the QRDR of *gyrA* and *parC*, which are defined as codons 67–106 of *gyrA* and codons 56–108 of *parC* [24]. All five nalidixic-acid-resistant *E. coli* isolates carried a Ser-83→Leu substitution in *gyrA*; the isolate which was also resistant to ciprofloxacin carried Asp-87→Asn substitution in *gyrA* as well as Ser-80→Ile substitution in *parC*. This is consistent with the view that a single mutation in *gyrA* can generate nalidixic acid resistance, but additional mutations in *gyrA* and/or *parC* are needed for high-level resistance to ciprofloxacin [24]. In addition to the mutations within the QRDR of *gyrA* and *parC*, mutations were identified outside of the QRDR. However, it is uncertain whether these mutations would contribute to resistance to quinolones and fluoroquinolones.

The most notable result from this study is the presence of *E. coli* belonging to ST131 in the surface water. ST131 is a globally disseminated clone that causes a wide range of infections, including urinary tract and bloodstream infections, in community and hospital settings [25]. Its multidrug resistance (MDR: resistant to two or more antimicrobial drugs), enhanced virulence, and rapid dissemination have made this strain of *E. coli* a significant public health threat worldwide [25]. The three ST131 isolates in this study were recovered from different seasons and in different locations; however, they shared many common characteristics. The three isolates were resistant to ampicillin and nalidixic acid, belonged to phylogenetic group B2, contained *bla*_TEM_, had mutations in *gyrA* and *parC*, and carried an IncF plasmid that was positive for the replicon FIB alone or along with the FII replicon. The shared resistance to ampicillin was likely mediated by *bla*_TEM_, which encodes TEM β-lactamase, usually located on an IncF plasmid [25]. In addition, reduced susceptibility to ciprofloxacin was seen in these three isolates, with the MICs ranging from 0.25 mg/L to >4 mg/L compared to other isolates from this study, whose MICs ranged from ≤0.015 mg/L to 0.03 mg/L. Only one of the three isolates was resistant to ciprofloxacin according to the Clinical Laboratory Standards Institute (CLSI) guideline (≥4.0 mg/L) [15]; however, more than a tenfold increase in the MICs of the other two isolates is significant as ciprofloxacin resistance has been commonly detected in ST131, limiting treatment options for infections caused by ST131 *E. coli* [25]. Moreover, fluoroquinolone-susceptible *E. coli* isolates of ST131 were identified in urinary tract infection patients, as well as in healthy populations [25]. In addition to the mutation in *gyrA* (Ser-83→Leu), the two ciprofloxacin-susceptible isolates also contained the Lys-247→Glu substitution in *parC*. This mutation outside the QRDR of *parC* could be responsible for the reduced susceptibility to ciprofloxacin in addition to the resistance to nalidixic acid in these isolates. All three ST131 isolates belonged to different pulsotypes, confirming the clonal diversity within ST131.

Of the three ST131 *E. coli* isolates, only one isolate encoded CTX-M-15. Among the diverse subclones of ST131, CTX-M-15-producing *E. coli* is the most widely distributed strain worldwide [26]. Since its first detection in India in 2001, this pandemic clone has been responsible for community and hospital-acquired infections [26]. Unlike the other two non-ESBL-producing ST131 isolates, the CTX-M-15 ESBL-producing ST131 was resistant to third-generation cephalosporins and had a reduced susceptibility to amoxicillin/clavulanic acid.

In addition to the ST131 *E. coli* isolates, an isolate positive for *bla*_CMY-2_ was identified. CMY-2 is the most commonly identified plasmid-mediated ampicillin (AmpC) β-lactamase across the world [27]. The *bla*_CMY-2_-positive isolate was resistant to a broad spectrum of β-lactams, but not inhibited by clavulanic acid, contained *bla*_TEM-1_, and exhibited resistance to multiple drugs (amoxicillin/clavulanic acid, ampicillin, cefoxitin, ceftiofur, ceftriaxone, and gentamicin). I1α and FIB replicon types were detected in the isolate, similar to the previous findings that *bla*_CMY-2_ is often present on Inc1 and IncF plasmids [28,29]. The isolate belonged to virulence-associated phylogenetic group B2 and corresponded to ST2552. The *bla*_CMY-2_ gene has often been detected in *E. coli* from food animals and their products throughout the world, which could be attributed to the agricultural use of expanded-spectrum cephalosporins in food animals [29,30].

Among the diverse STs seen in this study, the presence of ST10, ST58, ST69, ST86, ST88, and ST117 strains is notable. Even though these isolates were not ESBL producers, *E. coli* that belonged to these STs were present in other water sources, as well as human and animal clinical isolates as ESBL-producing *E. coli* in Netherlands and France [31,32]. Furthermore, the two ST69 isolates exhibited ampicillin resistance and contained *bla*_TEM_. ST10, ST58, and ST69 were each represented twice in this study, but the isolates of the same STs displayed different phenotypic and genotypic profiles. The three sets of matching isolates each belonged to the same phylogenetic group (ST10 to C, ST58 to B1, and ST69 to E), and contained IncF plasmids, but had different AR phenotypic profiles, AR genes, PFGE, and replicon type (RT) profiles. This demonstrates the diversity of *E. coli* in surface water.

MGEs, such as plasmids and integrons, are readily acquired by *E. coli* often conferring resistance in these bacteria [33]. These MGEs can be transferred between organisms of the same species or different genera, mediating the exchanges of AR genes in the environment. Our results showed that MGEs capable of such exchanges are present among 28 of the 34 AR waterborne *E. coli*. IncF plasmids were found in 20 isolates, nine of which carried more than one F replicon. FIB was the most common replicon, and was often detected in combination with FIA and/or FII replicons. Of nine *E. coli* isolates containing the TEM β-lactamase, eight isolates contained IncF plasmids, suggesting the possible carriage of *bla*_TEM_ on an IncF plasmid. Other plasmid types detected in this study were IncA/C, IncHI1, IncI1α, IncP, IncR, IncX1, and IncY. Among these plasmids, IncA/C, IncF, and IncI1 are reported to be highly associated with multiple resistance [34].

Integrons are also known to be associated with AR gene acquisition and dissemination; class 1 integrons are the most commonly identified with multiple resistance genes [33,35]. Integrons are not mobile by themselves but can be mobilized by other MGEs, such as transposons and plasmids, and can spread to other bacterial species [35]. Of the 15 MDR *E. coli* isolates from this study, three isolates carried class 1 integrons (20%; 3/15). This prevalence is comparable to studies conducted in the Rio Grande River in the Texas–Mexico region and Seine estuary in France, which identified class 1 integrons in 12.5% (4/32) and 8.9% (25/279) of the MDR *E. coli* isolated, respectively [36,37].

In this study, 34 AR *E. coli* isolates, none of which were clones, were recovered from 24 sampling sites from the Upper Oconee watershed near Athens, GA. Land uses draining to the sampling sites include areas of urban, suburban, exurban, and rural development, as well as industrial, agricultural, and medical facilities. The AR isolates and the AR genes seem to be randomly distributed around the watershed rather than being clustered in specific regions or only found in areas where there might be more usage of antimicrobial drugs. Several studies proposed municipal and hospital wastewaters as the main source of AR bacteria, including ESBL-producing *E. coli*, in aquatic environments [31,38]. Other studies suggested animal manure and feces from farm animals and wildlife may also be contributors, though to a lesser extent [22,31,39]. In this study, a CTX-M-15 ESBL-producing ST131 isolate was recovered from a sampling site downstream of a hospital (MIDO 826), a potential source of the pathogenic bacteria (Figure 3). Interestingly, ESBL-producing *E. coli* were not isolated at the sampling site immediately downstream of the hospital (MIDO 825). This suggests a more complex method of AR transference to *E. coli*. Studies conducted in Nebraska and South Carolina, USA, found that AR bacteria were also recovered in environments with low or no fecal or antimicrobial inputs [39,40]. This suggests the persistence and wide distribution of AR genes in the aquatic environment regardless of the degree of antibiotic selective pressure.

It is important to note that our methods did not use selection with antimicrobials to isolate *E. coli*, thus the 6.9% resistant isolates is most likely a true representation of the levels of AR *E. coli* in the streams near Athens, GA, which is not trivial. In separate on-going studies when selective pressure is applied with various antimicrobials during isolation, almost all water samples yielded resistant *E. coli* isolates (data not shown). The presence of AR bacteria in surface waters is a significant public health concern as surface water serves as a source of drinking water, is used for recreational activities, and is also used for irrigation of agricultural crops and water for food animals. Humans can be exposed to AR bacteria through contact with contaminated water sources. Likewise, livestock and wildlife animals can also be exposed through contact with contaminated water sources, further spreading these bacteria. Thus, aquatic environments serve as the interface, which brings humans, animals, and environments together, facilitating the spread of AR in between these compartments. 

## 5. Conclusions

The present study identified diverse strains of AR *E. coli* in surface waters, including an epidemic strain. A CTX-M-15 ESBL-producing ST131 *E. coli* was isolated, and to our best knowledge, this is the first report of the presence of this particular strain in surface water in the USA. AR *E. coli* can transfer their AR determinants to other bacteria, which can then be transferred to humans. The presence of MGEs, such as plasmids and integrons, in AR *E. coli* would increase the chances of AR gene dissemination in the environment. The identification of surface water as a reservoir of AR for human and veterinary antimicrobial drugs highlights the importance of continuous surveillance of AR bacteria in surface water.

## Figures and Tables

**Figure 1 ijerph-16-03761-f001:**
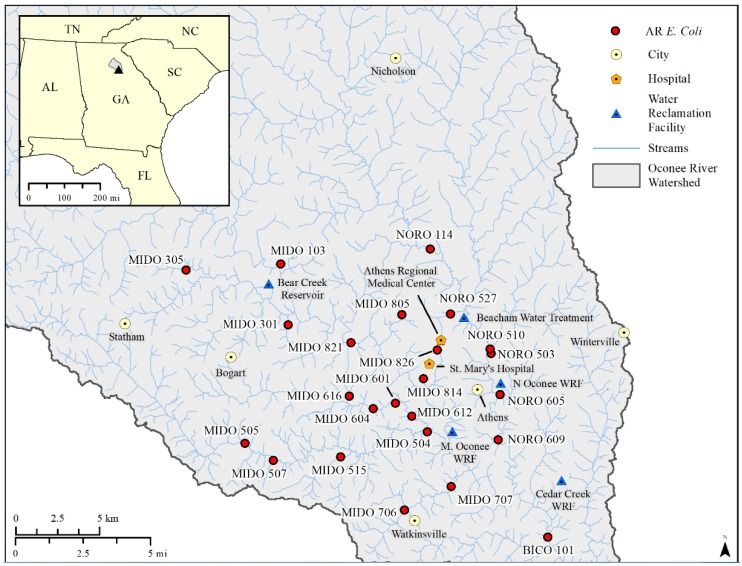
Map of water sampling sites in the Upper Oconee Watershed near Athens, GA. Sampling sites where antimicrobial resistant *E. coli* were isolated are labeled and are symbolized as red circles. Points of interest (circles—cities, triangles—water reclamation facilities, and pentagons—hospitals) are also labeled. The National Hydrography Dataset (NHD) streams are shown for reference. Inset map shows the Upper Oconee watershed in grey and Athens, GA, as a black triangle.

**Figure 2 ijerph-16-03761-f002:**
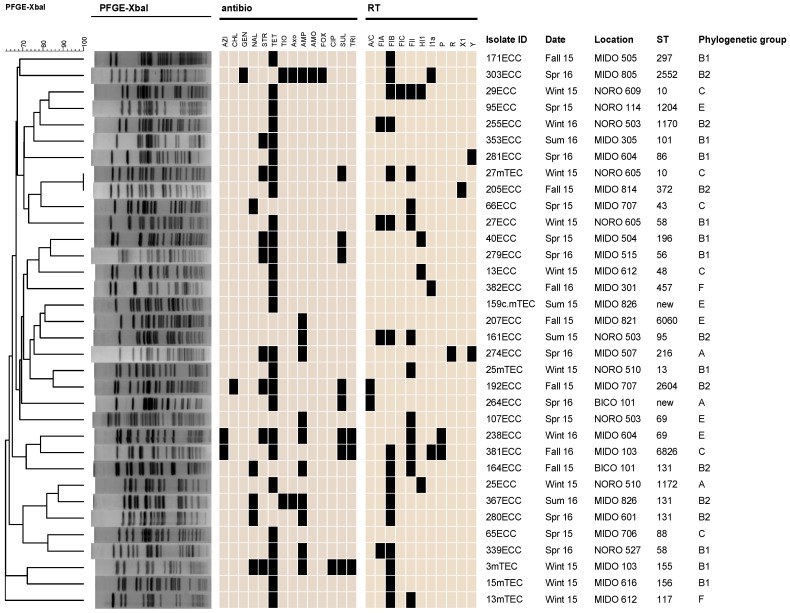
Dendrogram of 34 antimicrobial-resistant *E. coli* isolates recovered from Upper Oconee Watershed near Athens, GA. Their pulsed-field gel electrophoresis (PFGE) analysis, antimicrobial resistance patterns, replicon types, multilocus sequence typing (MLST), and phylogenetic groups are shown. Black boxes represent resistance to antimicrobials or the presence of plasmids, and beige boxes represent susceptibility to antimicrobials or absence of plasmids. Antimicrobials: azithromycin (AZI), chloramphenicol (CHL), gentamicin (GEN), nalidixic acid (NAL), streptomycin (STR), tetracycline (TET), ceftiofur (TIO), ceftriaxone (AXO), ampicillin (AMP), amoxicillin/clavulanic acid (AMO), cefoxitin (FOX), ciprofloxacin (CIP), sulfisoxazole (SUL), and trimethoprim/sulfamethoxazole (TRI).

**Figure 3 ijerph-16-03761-f003:**
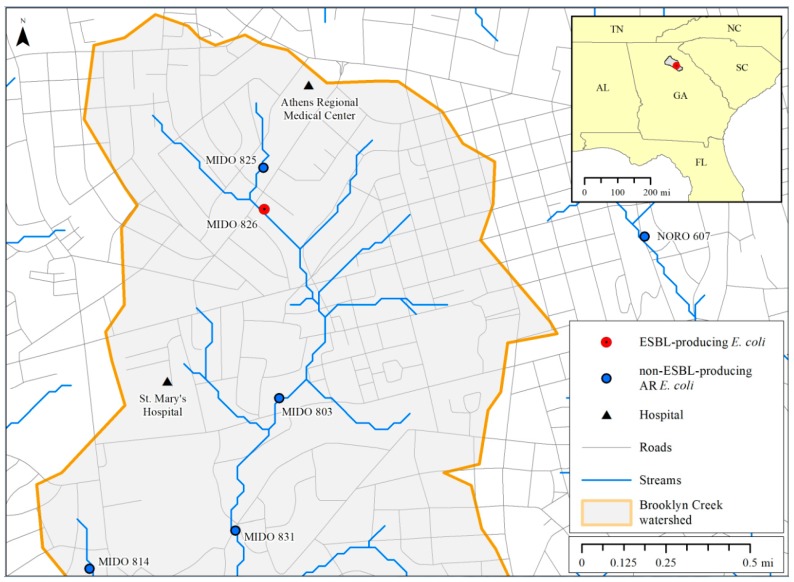
Map of water sampling sites in the Brooklyn Creek watershed in Athens, GA. Sampling site where extended-spectrum β-lactamase (ESBL)-producing *E. coli* was isolated (MIDO 826) is colored red. Sampling sites where Non-ESBL-producing *E. coli* were isolated are in blue. Hospitals are symbolized as black triangles.

**Table 1 ijerph-16-03761-t001:** Primers used for the identification of antimicrobial resistance genes and integron in resistant *E. coli* isolates from surface water.

Antimicrobial(s)/Integron	Target Gene	F Primer Sequence (5′ to 3′)	R Primer Sequence (5′ to 3′)	Amplicon Size (bp)	Reference
β-lactams	*bla* _CMY_	AACACACTGATTGCGTCTGA	GCCTCATCGTCAGTTATTGCA	1222	this study
	*bla* _CTX-M_	CACACGTGGAATTTAGGGACT	GAATGAGTTTCCCCATTCCGT	970	this study
	*bla* _TEM_	TTCTTGAAGACGAAAGGGC	ACGCTCAGTGGAACGAAAAC	1150	[6]
tetracycline	*tetA*	GCGCCTTTCCTTTGGGTTCT	CCACCCGTTCCACGTTGTTA	831	[7]
	*tetB*	CCCAGTGCTGTTGTTGTCAT	CCACCACCAGCCAATAAAAT	723	[7]
	*tetC*	TTGCGGGATATCGTCCATTC	CATGCCAACCCGTTCCATGT	1019	[7]
	*tetG*	AGCAGGTCGCTGGACACTAT	CGCGGTGTTCCACTGAAAAC	623	[7]
	*tetM*	GTGGACAAAGGTACAACGAG	CGGTAAAGTTCGTCACACAC	406	[8]
trimethoprim/	*dhfr1*	CGGTCGTAACACGTTCAAGT	CTGGGGATTTCAGGAAAGTA	220	[7]
sulphamethoxazole	*dhfr5*	CTGCAAAAGCGAAAAACGG	AGCAATAGTTAATGTTTGAGCTAAAG	432	[9]
	*dhfr12*	AAATTCCGGGTGAGCAGAAG	CCCGTTGACGGAATGGTTAG	429	[7]
	*dhfr13*	GCAGTCGCCCTAAAACAAAG	GATACGTGTGACAGCGTTGA	294	[7]
sulfisoxazole	*sul1*	TCACCGAGGACTCCTTCTTC	CAGTCCGCCTCAGCAATATC	331	[7]
	*sul2*	CCTGTTTCGTCCGACACAGA	GAAGCGCAGCCGCAATTCAT	435	[7]
	*cat1*	CTTGTCGCCTTGCGTATAAT	ATCCCAATGGCATCGTAAAG	508	[7]
	*cat2*	AACGGCATGATGAACCTGAA	ATCCCAATGGCATCGTAAAG	547	[7]
	*floR*	CTGAGGGTGTCGTCATCTAC	GCTCCGACAATGCTGACTAT	673	[7]
aminoglycosides	*aacC2*	GGCAATAACGGAGGCAATTCGA	CTCGATGGCGACCGAGCTTCA	450	[7]
	*aac*(3)*-IVa*	GATGGGCCACCTGGACTGAT	GCGCTCACAGCAGTGGTCAT	462	[7]
	*aadA1*	TATCAGAGGTAGTTGGCGTCAT	GTTCCATAGCGTTAAGGTTTCATT	484	[10]
	*aadA2*	TGTTGGTTACTGTGGCCGTA	GATCTCGCCTTTCACAAAGC	622	[10]
	*strA*	CTTGGTGATAACGGCAATTC	CCAATCGCAGATAGAAGGC	546	[11]
	*strB*	ATCGTCAAGGGATTGAAACC	GGATCGTAGAACATATTGGC	509	[11]
azithromycin	*mph*(A)	GTGAGGAGGAGCTTCGCGAG	TGCCGCAGGACTCGGAGGTC	403	[12]
ciprofloxacin,	*gyrA*	CGACCTTGCGAGAGAAAT	GTTCCATCAGCCCTTCAA	626	[13]
nalidixic acid	*parC*	AGCGCCTTGCGTACATGAAT	GTGGTAGCGAAGAGGTGGTT	965	[14]
class I integron	*intI1*	ACATGTGATGGCGACGCACGA	ATTTCTGTCCTGGCTGGCGA	568	[20]

**Table 2 ijerph-16-03761-t002:** Antimicrobial resistance genes detected in resistant *E. coli* isolates from surface water.

Resistance Phenotype (No. of Isolates Tested)	Resistance Gene Detected	No. of Resistance Gene Detected (%)
Ampicillin (n = 11)	*bla* _TEM-1_	9 (81.8)
Third generation cephalosporins	*bla* _CMY-2_	1 (50.0)
Ceftiofur, ceftriaxone (n = 2)	*bla* _CTX-M-15_	1 (50.0)
Azithromycin (n = 2)	*mph*(A)	2 (100.0)
Chloramphenicol (n = 1)	*floR*	1 (100.0)
Ciprofloxacin, nalidixic acid (n = 5)	*gyrA*	5 (100.0)
	*parC*	1 (20.0)
Streptomycin (n = 8)	*aadA1*	1 (12.5)
	*strA*	7 (87.5)
	*strB*	7 (87.5)
Sulfisoxazole (n = 8)	*sul1*	1 (12.5)
	*sul2*	7 (87.5)
Tetracycline (n = 26)	*tetA*	16 (61.5)
	*tetB*	9 (34.6)
	*tetC*	1 (3.8)
Trimethoprim/sulphamethoxazole (n = 3)	*dhfr5*	2 (66.7)
	*dhfr12*	1 (33.3)
Class I integron (n = 34)	*intI1*	3 (8.8)

**Table 3 ijerph-16-03761-t003:** Mutations in the *parC* and *gyrA* genes of the ciprofloxacin- and nalidixic-acid-resistant *E. coli* isolates from surface water.

Isolate ID	Resistance to (MIC in µg/mL)	*parC*	*gyrA*
3 mTEC	Ciprofloxacin (>4)	**Ser-80 → Ile**	Val-37 → Leu
	Nalidixic acid (>32)		**Ser-83 → Leu**
			**Asp-87 → Asn**
66 ECC	Nalidixic acid (>32)	No mutation	**Ser-83 → Leu**
164 ECC	Nalidixic acid (>32)	Lys-247 → Glu	**Ser-83 → Leu**
			Lys-162 → Gln
280 ECC	Nalidixic acid (>32)	Ala-192 → Gly	**Ser-83 → Leu**
367 ECC	Nalidixic acid (>32)	Lys-247 → Glu	**Ser-83 → Leu**

*Note*: Mutations in bold are those within the quinolone-resistance-determining regions (QRDR) of the *gyrA* and *parC* genes.

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
