# Peer review of "Genetic Characterization of Antimicrobial-Resistant Escherichia coli Isolated from a Mixed-Use Watershed in Northeast Georgia, USA"

_ijerph, 2019, doi:10.3390/ijerph16193761_

Round 1

Reviewer 1 Report

If it is allowed I would suggest writing the source/brand of DNA polymerase.  Not all enzymes available on market are DNA-free. Usually recombinant DNA polymerase is contaminated by DNA from E. coli used as a host for protein over-expression.

Author Response

If it is allowed I would suggest writing the source/brand of DNA polymerase. Not all enzymes available on market are DNA-free. Usually recombinant DNA polymerase is contaminated by DNA from coli used as a host for protein over-expression.

A sentence has been added at lines 84-85: “Taq DNA polymerase and PCR nucleotide mix used in PCR assays were purchased from Roche Diagnostics (Indianapolis, IN, USA).”

Reviewer 2 Report

AUTHORS

Manuscript ID: ijerph-612520

Title: Genetic characterization of antimicrobial resistant Escherichia coli isolated from a mixed-use watershed in Northeast Georgia, USA

Authors studied the role of surface water in the development and spread of antibiotic resistant bacteria. Its an interesting study that due to the novelty and quality of presented data, deserves to be published after minor concerns are addressed.

Please define MIC when first used

When describing PCR conditions, please disclose PCR Kit

On the PCR conditions, why to describe times and temperatures for just a few primer pairs? Why not all, or none?

On abstract authors state “Escherichia coli (n=496) were isolated, 34 (6.9%) of which were resistant to antimicrobials.”. But later authors state that they merely used the 34 isolates previously detected in a former study “The AR E. coli isolates (n=34) from a previous study were selected for characterization [5]”. This needs to be placed in the abstract for clarity

Author Response

Reviewer 2

Please define MIC when first used

Lines 111-113 now read: The “Isolates resistant to ceftriaxone (Minimum Inhibitory Concentration [MIC] ≥ 4 mg/L) were considered potential producers of extended-spectrum β-lactamase (ESBL) and selected for further testing.

When describing PCR conditions, please disclose PCR Kit

We did not use PCR kits and PCR for each gene was performed as previously described in the references. We have also added a sentence at lines 84-85 that reads: “Taq DNA polymerase and PCR nucleotide mix used in PCR assays were purchased from Roche Diagnostics (Indianapolis, IN, USA).”

On the PCR conditions, why to describe times and temperatures for just a few primer pairs? Why not all, or none?

For most of the PCR, we used the same PCR conditions as previously described in the references. However, for blaCTX-M and blaCMY, PCR conditions were described for the first time in this paper with the primers we used.

On abstract authors state “Escherichia coli (n=496) were isolated, 34 (6.9%) of which were resistant to antimicrobials.”. But later authors state that they merely used the 34 isolates previously detected in a former study “The AR E. coli isolates (n=34) from a previous study were selected for characterization [5]”. This needs to be placed in the abstract for clarity

Now the abstract reads: “In our previous study, 496 Escherichia coli were isolated from surface water, out of which 34 isolates were resistant to antimicrobials. For the current study, these 34 AR E. coli were characterized by pulsed-field gel electrophoresis, AR gene detection, plasmid replicon typing, class I integron detection, and multi-locus sequence typing.”

Reviewer 3 Report

The manuscript described genetic characterization of antimicrobial resistant (AMR) E. coli isolated from surface water from Georgia, US. Overall, the study was well designed and written and valued in understanding AMR gene determinants of environmental E. coli and potential associated risk. Minor comments are

Abstract, Line 17 was this already reported in the PloS One paper (Cho et al., 2018)?

Figure 1 has been published in the PloS One paper Cho et al., 2018)

Table 1, were the primers newly designed in the study or used from the published? If the latter, please provide reference. If all primers are published, the Table 1 can go supplemental.

Line 167, place [5] before comma

Author Response

Reviewer 3

Abstract, Line 17 was this already reported in the PloS One paper (Cho et al., 2018)?

We have made changes and now the abstract reads: “In our previous study, 496 Escherichia coli were isolated from surface water, out of which 34 isolates were resistant to antimicrobials. For the current study, these 34 AR E. coli were characterized by pulsed-field gel electrophoresis, AR gene detection, plasmid replicon typing, class I integron detection, and multi-locus sequence typing.”

Figure 1 has been published in the PloS One paper Cho et al., 2018)

In our previous paper, the figure contains the sampling sites for all our E. coli isolates but the Figure 1 in this paper contains sampling sites for only the resistant E. coli isolates. This is a more focused figure.

Table 1, were the primers newly designed in the study or used from the published? If the latter, please provide reference. If all primers are published, the Table 1 can go supplemental.

The references are provided in Table 1.

Line 167, place [5] before comma

The change has been made.